# A Best-Fitting B-Spline Neural Network Approach to the Prediction of Advection–Diffusion Physical Fields with Absorption and Source Terms

**DOI:** 10.3390/e26070577

**Published:** 2024-07-04

**Authors:** Xuedong Zhu, Jianhua Liu, Xiaohui Ao, Sen He, Lei Tao, Feng Gao

**Affiliations:** 1School of Mechanical Engineering, Beijing Institute of Technology, Beijing 100081, China; 3120205243@bit.edu.cn (X.Z.); jeffliu@bit.edu.cn (J.L.); hesen9712@163.com (S.H.); bit_taolei@163.com (L.T.); swai1213@163.com (F.G.); 2Hebei Key Laboratory of Intelligent Assembly and Detection Technology, Tangshan Research Institute, Beijing Institute of Technology, Tangshan 063000, China

**Keywords:** B-spline, best-fitting, physical fields, neural network, field gradient

## Abstract

This paper proposed a two-dimensional steady-state field prediction approach that combines B-spline functions and a fully connected neural network. In this approach, field data, which are determined by corresponding control vectors, are fitted by a selected B-spline function set, yielding the corresponding best-fitting weight vectors, and then a fully connected neural network is trained using those weight vectors and control vectors. The trained neural network first predicts a weight vector using a given control vector, and then the corresponding field can be restored via the selected B-spline set. This method was applied to learn and predict two-dimensional steady advection–diffusion physical fields with absorption and source terms, and its accuracy and performance were tested and verified by a series of numerical experiments with different B-spline sets, boundary conditions, field gradients, and field states. The proposed method was finally compared with a generative adversarial network (GAN) and a physics-informed neural network (PINN). The results indicated that the B-spline neural network could predict the tested physical fields well; the overall error can be reduced by expanding the selected B-spline set. Compared with GAN and PINN, the proposed method also presented the advantages of a high prediction accuracy, less demand for training data, and high training efficiency.

## 1. Introduction

At present, industry and scientific research extensively involve various forms of physical fields and their computation and simulation, such as flow field, temperature field, etc. The traditional way, which applies various numerical methods to solve these complex mathematical models, usually encounters problems related to complex methods, poor convergence, and the high cost in terms of time, and presents poor adaptability, especially for systems with strong nonlinearity and whose model parameters are difficult to determine [1,2,3,4,5,6]. In recent years, neural networks represented by deep learning have been successfully applied to the learning and prediction of many complex systems and scenarios, and applying machine learning to predict fields has become a hot topic [7,8,9]. However, a field has the characteristics of continuous spatial distribution, and field prediction needs to consider each spatial point of the huge amount of field data rather than one or more individual features. If a traditional fully connected neural network is directly used to predict the field, the number of nodes in the network output layer will be very large, making network training extremely difficult. By selecting an appropriate set of basis functions, fitting the huge amount of field data as low-dimensional vectors, and then training an ordinary neural network on the low-dimensional data, a better neural network model can be obtained [10,11]. At present, there exist some methods integrating neural networks and basis functions, such the B-spline neural network (BSNN), trigonometric-function neural network, and Walsh-function neural network [12,13].

Machine learning has a powerful ability to process complicated data. Its deep model hierarchical structure can automatically extract the inherent feature information of the original complex data and perform state recognition. Using machine-learning methods to predict spatially continuous fields has generated many concerns recently and the related challenges are still great. Convolutional neural networks (CNN) [14,15,16], generative adversarial networks (GAN) [17,18,19,20,21], and physics-informed neural networks (PINN) [22,23,24,25,26] are now widely used machine-learning methods to predict things with field features. As one of the most popular machine-learning methods, GAN is suited to predict the output field variables of complicated systems. It has been successfully utilized in many fields, including stress, deformation, and temperature fields, owing to its remarkable power in dealing with strongly nonlinear problems [27,28,29,30]. For the development of GAN, many novel GAN-based methods emerge, such as cGAN [18], StressGAN [31], and DCGAN [32]. PINN, a machine-learning method designed for the prediction of physical fields, has been successfully used to solve partial differential equations (PDEs) or PDE-based problems by adding data constraints and physics constraints in the loss function to constrain the space for admissible solutions. Benefiting from physics constraints, the PINN can reduce the requirement for labelled data and can be easily applied to a small data regime [33,34]. Xie et al. [35] predicted a 3D temperature field using a physics–data hybrid PINN model and presented a good predicted result. Zhu et al. [36] applied the PINN to predict the temperature and melt pool dynamics during metal AM processes.

For the learning and prediction problems of continuous physical systems, an ordinary neural network is trained by directly using the input and output data of the system, while a basis function–neural network combined method first fits the continuously distributed data using a selected function set and then obtains the weight vectors with small data, which are used to train the neural network ordinarily, finally obtaining the surrogate model of the physical system. The B-spline function has many advantages, such as local piecewise, which allows the function approximation to be adjusted locally; that is, the weight and parameter changes in a basis function only affect the local approximation, rather than the global. This feature of the B-spline function greatly enhances the stability of approximating complex functions [37] and can obtain a good fit for complex curves or surfaces [38]. A B-spline neural network learning a continuous system must learn a representing vector that can well approximate continuous data samples. Compared with directly using a fully connected neural network, BSNN is a local weight update method with the advantages of a fast convergence rate and low computational complexity and can greatly reduce the data size and training time [39,40]. Currently, BSNN has many successful applications in system identification and the prediction of zero-dimensional time series. Zhang et al. [41] used BSNN to approximate the nonlinear term of a satellite communication antenna system and compared it with a variety of traditional methods as well as radial basis function neural networks. Their results indicated that the BSNN has better decoupling and anti-interference abilities and predicts a more accurate and stable result. This is because B-spline functions have a good local adjustment feature, which leads to their good nonlinear approximation capabilities [42]. In recent years, the basis function neural network has had many other applications, and its superiority has been continuously verified.

The application of basis functions in neural network modeling is becoming more and more extensive, and the ability to predict physical fields using a neural network method is also on the way; however, the process of combining basis functions and neural networks to predict spatially continuous fields, which has great potential, still needs more research. This work proposed a best-fitting B-spline neural network (bBSNN). The process of fitting the physical field and minimizing the error draws on the coding process and best coding scheme in information theory. Its accuracy and performance were tested by a series of PDE numerical experiments with different vector sizes, field gradients, and field states, and verified through comparison with GAN and PINN. The effectiveness of the bBSNN was validated and the prediction error was analyzed and discussed.

## 2. Theory

### 2.1. Best-Fitting B-Spline Function

A one-dimensional B-spline function set with an order of *k* and an element number of *m* can be defined as follows:(1)$$\left\{\begin{array}{l} N_{j}^{k}\left(x\right)=\frac{x-t_{j}}{t_{j+k}-t_{j}}N_{j}^{k-1}\left(x\right)+\frac{t_{j+k+1}-x}{t_{j+k+1}-t_{j+1}}N_{j+1}^{k-1}\left(x\right)\\ N_{j}^{0}\left(x\right)=\left\{\begin{array}{ll} 1, & t_{j}\leq x\leq t_{j+1}\\ 0, & elsewhere \end{array}\right. \end{array}\right.$$
where {*t_j_*} is a knot sequence, {*t_j_*|*j* = 1, 2, …, *m* + *k* + 1}. Assuming that *x* goes from *x*_min_ to *x*_max_, the knot sequence for a uniform B-spline function can then be provided in ascending order as follows:(2)$$t_{1}<t_{2}<\ldots t_{k+1}=x_{\min}<t_{k+2}<\ldots <t_{m}<x_{\max}=t_{m+1}<\ldots <t_{m+k+1}$$
Here, *t_j_* (*j* < *k* + 1 or *j* > *m* + 1) are external knots, *t_j_* (*j* > *k* + 1 or *j* < *m* + 1) are internal knots, and the other two are end knots. For a higher-dimension B-spline function, all the one-dimensional functions defined in each dimension need to be multiplied, as in Equation (1). Giving a two-dimensional function *ϕ*(*x*, *y*), we can have approximation *ϕ**(*x*, *y*) composed of a linear combination of B-splines and a weight vectors **a,** as follows:(3)$$\phi^{*}\left(x,y\right)=\sum_{l=1}^{m_{y}}\sum_{j=1}^{m_{x}}a_{lj}N_{j}^{k_{x}}\left(x\right)N_{l}^{k_{y}}\left(y\right)=N{\left(x,y\right)}^{T}a$$
Here, *m_x_* and *m_y_* are the numbers of B-spline functions in the *x* and *y* dimensions, respectively, and *k_x_* and *k_y_* are their corresponding orders. Rewriting this formula in terms of matrices, we have:(4)$$N\left(x,y\right)={\left\{\left[N_{11}\cdots N_{1j}\cdots N_{1m_{x}}\right]\cdots \left[N_{l1}\cdots N_{lj}\cdots N_{lm_{x}}\right]\cdots \left[N_{m_{y}1}\cdots N_{m_{y}j}\cdots N_{m_{y}m_{x}}\right]\right\}}^{T}$$
and
(5)$$a={\left\{\left[a_{11}\cdots a_{1j}\cdots a_{1m_{x}}\right]\cdots \left[a_{l1}\cdots a_{lj}\cdots a_{lm_{x}}\right]\cdots \left[a_{m_{y}1}\cdots a_{m_{y}j}\cdots a_{m_{y}m_{x}}\right]\right\}}^{T}$$
where $N_{lj}=N_{j}^{k_{x}}\left(x\right)N_{l}^{k_{y}}\left(y\right)$. In practice, function *ϕ*(*x*, *y*) is often given by *n_x_* × *n_y_* discrete points, *ϕ*(*x_i_*, *y_p_*), for which we have a fitting error:(6)$$E=\frac{1}{2}\sum_{p=1}^{n_{y}}\sum_{i=1}^{n_{x}}{\left[\phi \left(x_{i},y_{p}\right)-\phi^{*}\left(x_{i},y_{p}\right)\right]}^{2}$$
Applying the least square method to minimize the fitting error *E*, we have a best-fitting weight vector **a**:(7)$$a={\left[A^{T}A\right]}^{-1}\left[A^{T}B\right]$$
Here
(8)$$A={\left\{\left[N\left(x_{1},y_{1}\right)\cdots N\left(x_{i},y_{1}\right)\cdots N\left(x_{n_{x}},y_{1}\right)\right]\cdots \left[N\left(x_{1},y_{p}\right)\cdots N\left(x_{i},y_{p}\right)\cdots N\left(x_{n_{x}},y_{p}\right)\right]\cdots \left[N\left(x_{1},y_{n_{y}}\right)\cdots N\left(x_{i},y_{n_{y}}\right)\cdots N\left(x_{n_{x}},y_{n_{y}}\right)\right]\right\}}^{T}$$
and
(9)$$B={\left\{\left[\phi \left(x_{1},y_{1}\right)\cdots \phi \left(x_{i},y_{1}\right)\cdots \phi \left(x_{n_{x}},y_{1}\right)\right]\cdots \left[\phi \left(x_{1},y_{p}\right)\cdots \phi \left(x_{i},y_{p}\right)\cdots \phi \left(x_{n_{x}},y_{p}\right)\right]\cdots \left[\phi \left(x_{1},y_{n_{y}}\right)\cdots \phi \left(x_{i},y_{n_{y}}\right)\cdots \phi \left(x_{n_{x}},y_{n_{y}}\right)\right]\right\}}^{T}$$

By using the best fitting, a function *ϕ*(*x*, *y*) or its discrete version *ϕ**(*x*, *y*) can be represented by a weight vector and a pre-selected B-spline function set; as a consequence, a data set including amounts of *ϕ**(*x*, *y*) can be degraded into a series of corresponding weight vectors along with a known B-spline function set. In general, the scale of *ϕ**(*x*, *y*) is far greater than that of the corresponding weight vector, which is determined by the number of B-splines in the selected function set. Additionally, B-spline fitting can filter the high-frequency modes in *ϕ**(*x*, *y*); the extent of this can be modified by changing the number or the order of the B-splines in the function set. Since the B-spline function is piecewise, the B-spline fitting can well approximate the high-frequency modes through the addition of more low-order B-splines to the function set instead of higher-order basis functions.

### 2.2. Combination of Neural Network and B-Spline Function

The architecture of the bBSNN is shown in Figure 1, which combines a fully connected neural network and the B-spline function fitting. A data set {OUT: *ϕ*(*x*, *y*)} with *r* samples along with *r* corresponding control vectors {IN: **p**} can be used as an example.
(10)$$\begin{array}{l} \left\{\mathrm{In}:p\right\}=\left\{p_{1},p_{2},\cdots ,p_{r}\right\}\\ \left\{\mathrm{Out}:\phi \left(x,y\right)\right\}=\left\{\phi_{1,1},\phi_{1,2},\cdots ,\phi_{n_{x},n_{y}}\right\} \end{array}$$
The training and prediction of this method can be illustrated as follows (Please refer to the Appendix A for the bBSNN code.):(1)Use the numbers *m_x_* and *m_y_* and the orders *k_x_* and *k_y_*, and calculate the knot sequence in each dimension according to the domain boundaries of the data set, then generate the B-spline function set;(2)Calculate the best-fitting weight vector **a** using the least square method; thus, the data set can be replaced by:
(11)$$\left\{\mathrm{Out}:\phi \left(x,y\right)\right\}=\left\{a_{1,1},a_{1,2},\cdots ,a_{m_{x},m_{y}}\right\}$$(3)Train an ordinary fully connected neural network whose input and output are now as follows:
(12)$$\begin{array}{l} \left\{\mathrm{In}:p\right\}=\left\{p_{1},p_{2},\cdots ,p_{r}\right\}\\ \left\{\mathrm{Out}:a\right\}=\left\{a_{1,1},a_{1,2},\cdots ,a_{m_{x},m_{y}}\right\} \end{array}$$(4)Once the neural network is well trained, a new control vector **p** can be used to predict a new weight vector **a**; predicted function *ϕ**(*x*, *y*) is then restored by Equation (3).

## 3. Numerical Experiments and Discussions

In this section, we focus on the numerical experiments conducted on general advection–diffusion physical fields, using absorption and source terms to examine the performance of the bBSNN, in which the boundary conditions and the scalar field are regarded as the input control vector {IN: **p**} and the output data {OUT: *ϕ*(*x*, *y*)}, respectively. Here, we consider the physical fields as follows:(13)$$a\nabla^{2}\phi -b\nabla \cdot \left(u\phi \right)-c\phi +d=0$$
where *ϕ* is a scalar field and **u** is a vector field, such as the velocity field. *a*, *b*, *c*, and *d* are the coefficients of the diffusion, convection, absorption, and source terms, respectively. In the numerical experiments, a vortical velocity field with an average velocity of 1 is implanted into the PDE and expressed as follows:(14)$$u=\left[\begin{array}{l} u\left(x,y\right)\\ v\left(x,y\right) \end{array}\right]=2\left[\begin{array}{cc} 0 & 1\\ -1 & 0 \end{array}\right]\left[\begin{array}{l} x/\left(0.5L_{x}\right)-1\\ y/\left(0.5L_{y}\right)-1 \end{array}\right]$$
Here, *L_x_* and *L_y_* are the dimensions of the domain; *u* = *u*(*x*, *y*) and *v* = *v*(*x*, *y*) are the velocity components. The physical fields can be characterized by three parameters: (1) *α* = *a*/*b*, representing the diffusion rate of the scalar field; (2) Pécklet number *Pe* = *LU*/*α*, a dimensionless number representing the ratio of the convection intensity to the diffusion intensity of the scalar field; (3) *My* = *L*^2^*C*/*a*, a dimensionless number representing the ratio of the absorption intensity to the diffusion intensity of the scalar field. Here, *L*, *U*, and *C* are the characteristic length, velocity, and absorption rate of the system, respectively. (Please refer to the Appendix A for the finite volume method calculation code of the above physical field.)

### 3.1. Numerical Experiment Setup and Preliminary Verification

For the numerical experiment setup, a square domain is configured for the physical fields, and two independent and variable conditions and two fixed conditions are set at the four boundaries of the domain for preliminary verification, as shown in Figure 2, yielding an input control vector **p** = [*B*_1_, *B*_2_]^T^, whose values are slightly smoothed in the vicinity of their junction points by a hyperbolic tangent function. The parameters of the physical fields are given as *a* = 10, *b* = 200, and *c* = 0.5, *d* = 1000, which yields a field state with *α* = 0.05, *Pe* = 20, and *My* = 0.05. The numerical result of the scalar field *ϕ*(*x*, *y*) is obtained with a grid resolution of 128 × 128 using a finite volume method. The configuration of the B-spline function set is *k_x_* = *k_y_* = 2 and *m_x_* = *m_y_* = 10, determining a weight vector **a** with a size of 100 × 1.

For the preliminary tests and verification, we generated a data set with 100 samples to train the neural network, where the input control vectors **p** were randomly generated and *B*_1_, *B*_2_ ∈ [0, 100]. By computing the DSSA physical fields numerically, 100 corresponding field data *ϕ*(*x*, *y*) can be obtained, which yields 100 corresponding weight vectors **a** when using the best fit and the selected B-spline function set. Three hidden layers with 8, 10, and 12 nodes, respectively, and one output layer were configured for the fully connected neural network, while two S-shaped tangent functions and two pure linear functions were selected for the corresponding layers. After the network was trained, we verified the bBSNN method via predicting a case with [*B*_1_ *B*_2_]^T^ = [80 50]^T^, which is a new sample that has not appeared in the training data set. Figure 3 shows the ground truth and predicted fields of this case, as well as the error between them. To evaluate the error field, we calculated the averaged value *μ*, standard deviation *σ*, and the averaged absolute error *μ*_|*ϕ**−*ϕ*|_. When comparing Figure 3a with Figure 3b, they are shown to be extremely similar. Figure 3c shows the error field, and it can be seen that most of the errors are small, except those reaching about 10 at the corners or close to the boundaries. The average error *μ* = 0.0131, the standard deviation *σ* =1.4080, and the averaged absolute error *μ*_|*ϕ**−*ϕ*|_ = 0.6952. This quantitatively indicates that the agreement between the prediction and the ground truth is good.

Qualitatively, it can be seen from Figure 3 that where the gradient is large, the error is likely to be large. To further reduce the error, we added more B-splines to the function set to increase the resolution of the high-frequency components of the field when finding the best fitting. Figure 4a shows the prediction results when using the B-spline function set with *k_x_* = *k_y_* = 3 and *m_x_* = *m_y_* = 15. The error field in Figure 4b shows that the averaged error *μ* = −0.0012, the standard deviation *σ* = 0.3589, the averaged absolute error *μ*_|*ϕ**−*ϕ*|_ = 0.1160, and the maximum error is about 6. This indicates that the error can be greatly reduced by increasing the number of functions of the selected B-spline set. However, the computational cost and complexity of the training are also greatly increased.

### 3.2. Effect of the Size of the Input Control Vectors

To examine the performance of the bBSNN method when predicting more complicated DSSA physical fields, we add more independent and variable boundary conditions to expand the size of the input control vectors. In this subsection, two numerical experiments are carried out, where the configuration of the selected B-spline function set is still so that *k_x_* = *k_y_* = 2 and *m_x_* = *m_y_* = 10, while the independent and variable boundary conditions are increased to 4 and 6, respectively. The data sets of these two numerical experiments are given 150 and 200 random samples, respectively, with the control vector *B_i_* ∈ [0, 100]. For the architecture of the fully connected neural network, we only change the nodes of the input layer to match the size of the control vector and keep the hidden layers, the output layer, and the activation functions the same as those in Section 3.1.

Figure 5a shows the configuration of the numerical experiment with four independent boundary conditions, and the DSSA physical fields with the same parameters used in Section 3.1 were still computed at a grid resolution of 128 × 128, obtaining the ground-truth field under a tested condition [*B*_1_ *B*_2_ *B*_3_ *B*_4_]^T^ = [80 50 25 60]^T^, as shown in Figure 5b. After the neural network was well trained, the predicted field obtained under the same conditions by the bBSNN is shown in Figure 5c, which is also very close to the ground-truth field. Figure 5d shows the error field, where *μ* = −0.0061, *σ* = 0.8270, *μ*_|*ϕ**−*ϕ*|_ = 0.3991, and the maximum error is about 12. Compared with the experiment with two independent boundary conditions, the characteristics of the error field are similar, while the error statistics are even smaller. This is likely due to the smaller field gradient under the tested condition.

Similarly, Figure 6 shows the configuration of the numerical experiment with six independent boundary conditions and the ground-truth field, as well as that predicted under a tested boundary condition [*B*_1_ *B*_2_ *B*_3_ *B*_4_ *B*_5_ *B*_6_]^T^ = [80 50 25 60 100 40]^T^. Figure 6d shows the error field, where *μ* = −0.0138, *σ* = 0.9471, *μ*_|*ϕ***−ϕ*|_ = 0.4792. Compared with the experiment with four independent boundary conditions, the error slightly increases, while it is still less than that of the experiment with two. These numerical experiments indicate that the size of the input control vector has little effect on the accuracy of the bBSNN in predicting the steady-state DSSA physical fields, and the error variations are mainly due to the differences in the local field gradients.

### 3.3. Effect of the Field Gradient

The aforementioned numerical experiments also imply a potential great influence from the field gradient on the accuracy of the bBSNN method. To further examine the effect of the field gradient, we change the value range of the boundary conditions to generate two data sets with greatly different field gradients in the physical fields with two independent boundary conditions, where *B*_1_, *B*_2_ ∈ [0, 300] and *B*_1_, *B*_2_ ∈ [0, 900], respectively. Additionally, the configuration of the domain, the selected B-spline function set (*k_x_* = *k_y_* = 3 and *m_x_* = *m_y_* = 15), and the architecture of the fully connected neural network are the same as those used in the second test in Section 3.1. For these two data sets, we randomly generated 150 samples in their corresponding ranges. After these two networks were trained, the cases with conditions [*B*_1_ *B*_2_]^T^ = [240 150]^T^ and [720 450]^T^ were predicted, respectively. Figure 7a–c show the ground-truth field, predicted field, and the error field under the tested condition [*B*_1_ *B*_2_]^T^ = [240 150]^T^, respectively, where *μ* = 0.0103, *σ* = 1.0766, and *μ*_|*ϕ**−*ϕ*|_ = 0.3525, and the maximum error is about 18. Figure 7d–f show the corresponding results under the tested conditions [*B*_1_ *B*_2_]^T^ = [720 450]^T^, where *μ* = 0.0254, *σ* = 3.2288 and *μ*_|*ϕ**−*ϕ*|_ = 1.0429, and the maximum error is about 55. It can be seen that the error fields of these two cases are fairly similar except for their exact values. These results indicate that the prediction error of the bBSNN method is sensitive to the local field gradient. Specifically, comparing these two cases and the second case in Section 3.1, where *B*_1_, *B*_2_ ∈ [0, 100], we can further conclude that the error of the bBSNN when predicting the steady-state DSSA physical fields is approximately proportional to the field gradient.

### 3.4. Effect of the Field State

In this subsection, we further examine the performance of the bBSNN in predicting the physical fields with various field states, which are governed by the coefficients *a*, *b*, *c*, and *d* of Equation (13). Here, we carry out two representative numerical experiments: (1) *a* = 1, *b* = 1, *c* = 1, and *d* = 1, yielding the system parameters *α* = 1, *Pe* = 1, and *My* = 1; (2) *a* = 1, *b* = 100, *c* = 10, and *d* = 10, yielding the system parameters *α* = 0.01, *Pe* = 100, *My* = 10. The system parameters indicate that the conditions are moderate for the first experiment while they are much more severe for the second one. Specifically, the second experiment will suffer from a much lower diffusion rate alongside much stronger advection and absorption effects, resulting in much higher local field gradients. Additionally, the domain configuration, B-spline function set (*k_x_* = *k_y_* = 3 and *m_x_* = *m_y_* = 15), and the architecture of the fully connected neural network are kept the same as those used in the second test in Section 3.1, where 100 samples are used for each data set to train the network. The tested conditions for these two numerical experiments are both [*B*_1_ *B*_2_]^T^ = [80 50]^T^. Their ground-truth and predicted results, as well as the error fields, are shown in Figure 8. It can be seen that the pattern agreements between the predicted and the ground-truth fields for these two experiments are very good; however, the patterns of the error field are quite different, although the maximum errors are very close to each other. We calculated the error statistics for these: *μ* = 0.0069, *σ* = 0.2960, *μ*_|*ϕ**−*ϕ*|_ = 0.0781 for the first one and *μ* = 0.5788, *σ* = 1.0534, *μ*_|*ϕ**−*ϕ*|_ = 0.9219 for the second one. This indicates that the error ranges for these two experiments with different field states are similar, while the error statistics of the experiment with severe conditions are much larger than those of the one with moderate conditions.

### 3.5. Comparison with Analytical Solutions

The general form of the two-dimensional diffusion equation is as follows:(15)$$\frac{\partial \phi }{\partial t}=D\left(\frac{\partial^{2}\phi }{\partial x^{2}}+\frac{\partial^{2}\phi }{\partial y^{2}}\right)$$
where *Φ* is the scalar field and *D* is the diffusion coefficient. If the above-mentioned diffusion equation has an analytical solution, it usually has specific boundary conditions and initial conditions. Therefore, we assume a case where there is an instantaneous point diffusion source in the region, as shown in Figure 3, and the initial conditions are as follows:(16)$$\phi (x,y,0)=50\delta (x-x_{0},y-y_{0})$$
Here, *δ* is the Dirichlet function. Under such conditions, there exists an analytical solution for Equation (15), as follows:(17)$$\phi (x,y,t)=\frac{50}{4\pi Dt}e^{\left(-\frac{{\left(x-x_{0}\right)}^{2}+{\left(y-y_{0}\right)}^{2}}{4Dt}\right)}$$
Set *D* = 0.2, take the field distribution at *t* = 1 s as the target, take any point in the region shown in Figure 3 as the diffusion source, and record its coordinates (*x*_0_, *y*_0_) as the control vector. Randomly generate 100 sets of control vectors and their corresponding field distributions as training data sets for the bBSNN, where the network structure and parameter settings are the same as in Section 3.1. Finally, the prediction effect is verified with the diffusion source coordinates of (0.7, 0.8) and the verification results are shown in Figure 9. It can be seen from Figure 9a,b that the bBSNN prediction performance is good, the ground truth and prediction are extremely similar, and the error distribution is very average. The average error *μ* = 0.0187, the standard deviation *σ* = 0.0471, and the averaged absolute error *μ*_|*ϕ**−*ϕ*|_ = 0.0398. This shows that the bBSNN also shows great potential for predicting the diffusion field distribution with analytical solutions.

### 3.6. Method Comparision

To further demonstrate the effectiveness of bBSNN in predicting diffusion–advection–absorption–source physical fields, we compare it with GAN [18] and PINN [25]. The network structure of GAN is set up as shown in Figure 10a, mainly comprising a generator and discriminator. The generator is composed of seven transposed convolutional layers, and the discriminator is composed of six convolutional layers and one fully connected layer. The network structure of PINN is shown in Figure 10b, where the loss function includes three parts: partial differential structure loss (PDE loss), boundary value condition loss (BC loss), and real data condition loss (data loss), and the hidden layer is composed of six fully connected layers. The numerical example presented in Section 3.1 is used to train the GAN and PINN, and the boundary condition of the test group is also set as [*B*_1_ *B*_2_]^T^ = [80 50]^T^.

All the numerical cases were computed on PYTHON 3.6 with PYTORCH 1.10 and a computer with the configurations of Intel Xeon E5-2678@2.50 GHz and an RAM of 128 GB. To compare the performances of the bBSNN, GAN, and PINN, we summarized the training data set, training time, predicted results, and errors, as listed in Table 1. Notably, for the presented numerical problem, the bBSNN can be well trained using a data set with 150 images; however, the GAN cannot provide a converged result when the training data set is smaller than 1500, and we thus trained the GAN using a data set with 2000 images. Comparing their training times, the bBSNN has a much shorter training time than the GAN and PINN. Comparing their error maps, both the absolute averaged error |*μ*| and the averaged absolute error *μ*_|*ϕ**−*ϕ*|_ of the bBSNN are the smallest. These indicate that the training efficiency and the prediction accuracy of bBSNN are both much better than those of GAN and PINN, at least for the presented PDE problem.

## 4. Conclusions

This work introduced a best-fitting B-spline neural network, which combines a fully connected neural network and a best-fitting B-spline function set. Especially, we carried out a numerical study on the bBSNN method predicting steady-state advection–diffusion physical fields with absorption and source terms, from which the performance of the bBSNN was verified, and the effects of the physical fields and B-spline function set on the error field were investigated and discussed in detail. The numerical experiments indicated that the bBSNN method could predict the physical fields very well. From the error analysis of the numerical experiments, we can conclude the following:(1)The error of the bBSNN is sensitive to the local field gradient, and where the gradient is higher, the error is likely larger, almost showing a proportional relationship. This can be understood as meaning that, in information theory, when a signal or piece of data has certain characteristics, its information entropy may change. There is a certain correspondence between the sensitivity of the error and the variability of information entropy.(2)The effect of the field state slightly affects the range of the error field while greatly increasing the error statistics.(3)The prediction error of the bBSNN method can be reduced by increasing the order or the number of the B-splines in the selected function set.(4)The data set used to train the bBSNN can be very small, even for the case with six independent boundary conditions, the bBSNN trained by a data set with only 200 random samples can yield a good predicted field, and the training efficiency is also very high.(5)Compared with GAN and PINN, bBSNN presents obvious advantages in terms of training efficiency and prediction accuracy.

Consequently, we believe the bBSNN method can be used as a good surrogate model to predict advection–diffusion physical fields under relatively complex conditions, and could also be used to accelerate the numerical computation of such physical fields. Although this work is carried out for steady-state advection–diffusion physical fields, the bBSNN method, if combined with a recurrent neural network, may have great potential to deal with transient physical fields, which could be our next step in the near future.

## Figures and Tables

**Figure 1 entropy-26-00577-f001:**
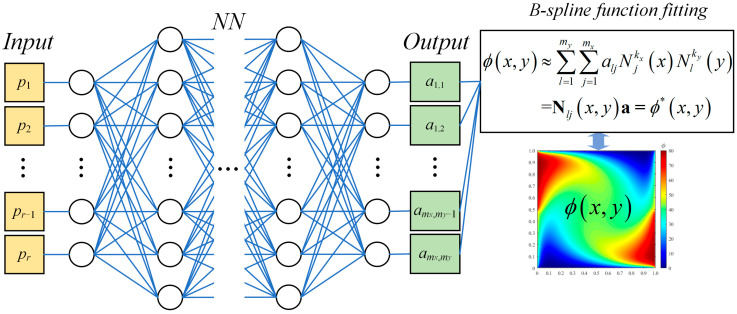
A schematic of the architecture of the best-fitting B−spline neural network.

**Figure 2 entropy-26-00577-f002:**
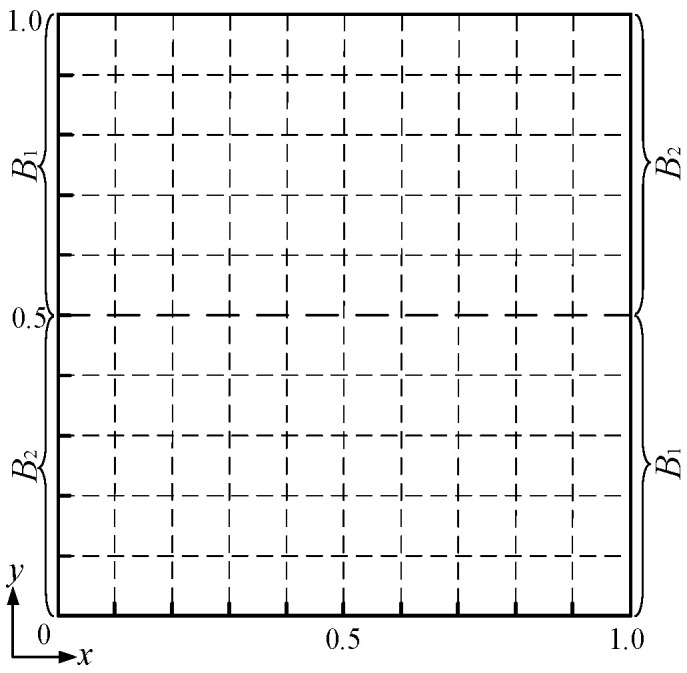
Configurations of the numerical experiments.

**Figure 3 entropy-26-00577-f003:**
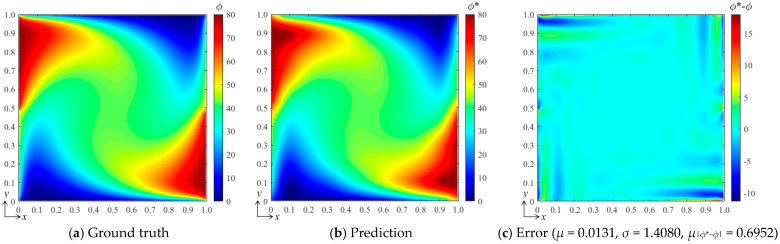
(**a**) Ground-truth steady-state field obtained by computing the DSSA physical fields using the finite volume method on a grid resolution of 128 × 128; (**b**) result predicted by the bBSNN with second-order 10 × 10 functions; (**c**) the error field between the ground truth and the prediction, where *μ*, *σ*, and *μ*_|*ϕ**−*ϕ*|_ are the averaged error, standard deviation, and the averaged absolute error, respectively.

**Figure 4 entropy-26-00577-f004:**
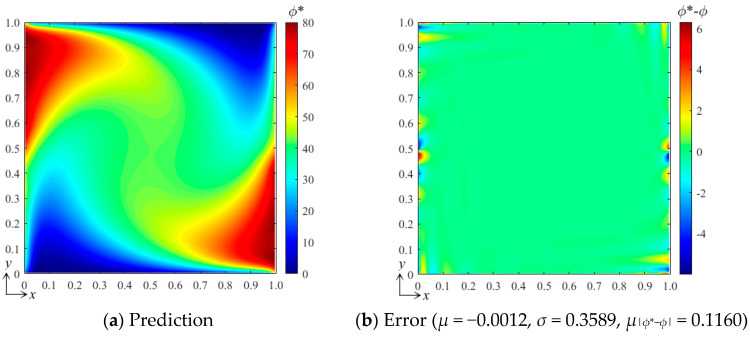
(**a**) The result predicted by the bBSNN with the second-order 15 × 15 functions and (**b**) the error field.

**Figure 5 entropy-26-00577-f005:**
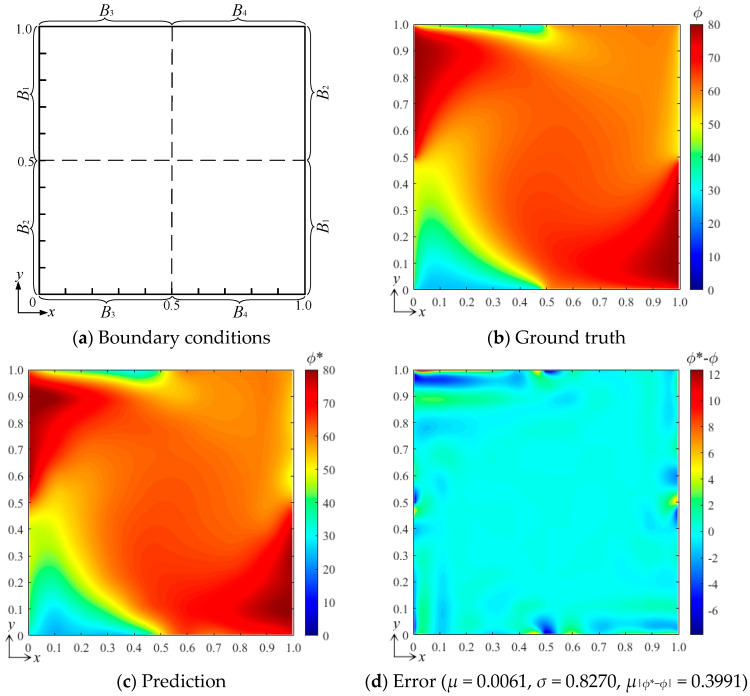
(**a**) The configuration of the numerical experiment with four independent boundary conditions, (**b**) the ground-truth field under the tested conditions [*B*_1_ *B*_2_ *B*_3_ *B*_4_]^T^ = [80 50 25 60]^T^, (**c**) the predicted field, and (**d**) the error field of steady-state DSSA physical fields.

**Figure 6 entropy-26-00577-f006:**
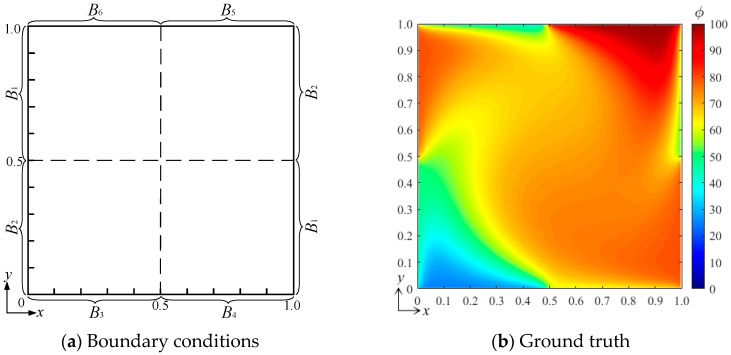
(**a**) The configuration of the numerical experiment with six independent boundary conditions, (**b**) the ground-truth field under the tested conditions [*B*_1_ *B*_2_ *B*_3_ *B*_4_ *B*_5_ *B*_6_]^T^ = [80 50 25 60 100 40]^T^, (**c**) the predicted field, and (**d**) the error field of a steady-state DSSA physical fields.

**Figure 7 entropy-26-00577-f007:**
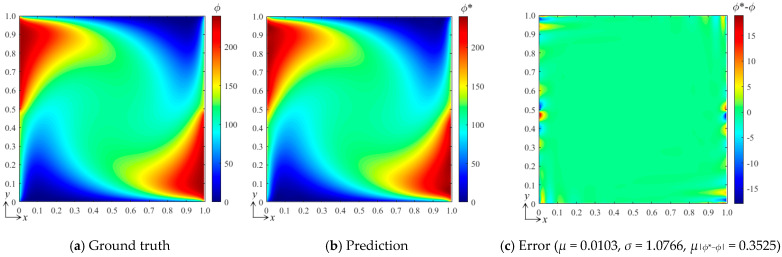
(**a**) The ground-truth field under the tested condition [*B*_1_ *B*_2_]^T^ = [240 150]^T^, (**b**) the predicted field by the bBSNN trained with a data set generated from *B*_1_ and *B*_2_ ∈ [0, 300], and (**c**) the error field; (**d**) the ground-truth, (**e**) predicted, and (**f**) error fields under the tested condition [*B*_1_ *B*_2_]^T^ = [720 450]^T^ and *B*_1_, *B*_2_ ∈ [0, 900].

**Figure 8 entropy-26-00577-f008:**
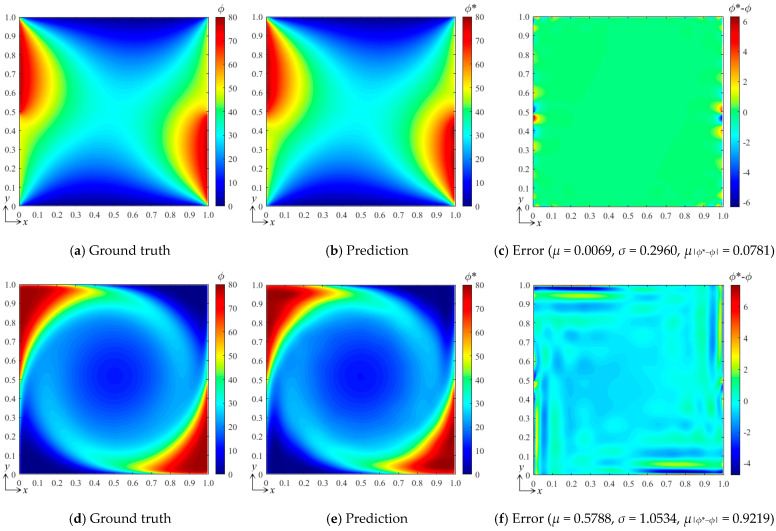
(**a**) The ground-truth field, (**b**) the predicted field, and (**c**) the error field of the numerical experiment with *a* = 1, *b* = 1, *c* = 1, and *d* = 1, leading to *α* = 1, *Pe* = 1, and *My* = 1 for the DSSA physical fields; (**d**) the ground-truth, (**e**) predicted, and (**f**) error fields of the numerical experiment with *a* = 1, *b* = 100, *c* = 10, and *d* = 10, leading to *α* = 0.01, *Pe* = 100, and *My* = 10.

**Figure 9 entropy-26-00577-f009:**
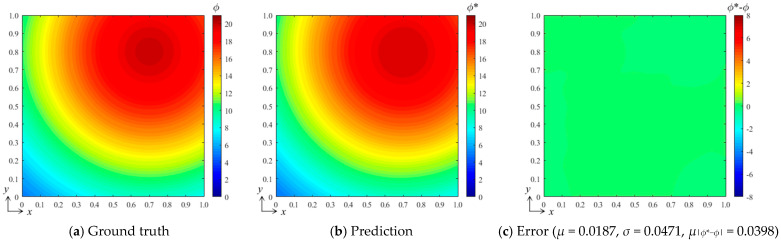
(**a**) Ground truth obtained by computing the diffusion fields using the analytical solutions; (**b**) the result predicted by the bBSNN with the third-order 15 × 15 functions; (**c**) the error field between the ground truth and the prediction, where *μ*, *σ*, and *μ*_|*ϕ**−*ϕ*|_ are the averaged error, standard deviation, and the averaged absolute error, respectively.

**Figure 10 entropy-26-00577-f010:**
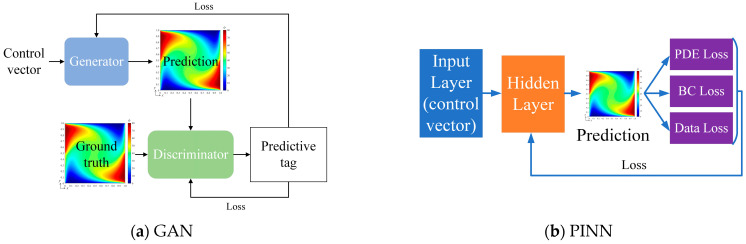
Schematics of the architectures of the used GAN and PINN.

**Table 1 entropy-26-00577-t001:** The comparison of bBSNN, GAN, and PINN.

	bBSNN	GAN	PINN
Training data set	150	2000 (<1500 non-converged)	-
Training time	5 min	290 min	15 min
Prediction	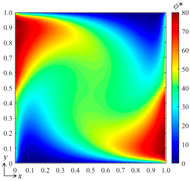	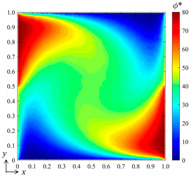	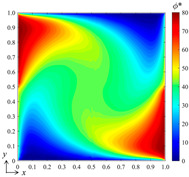
Error map	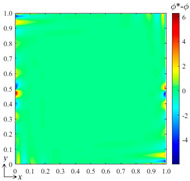	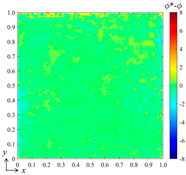	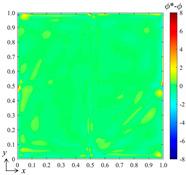
Error	*μ* = −0.0012, *σ* = 0.3589, *μ*_|*ϕ**−*ϕ*|_ = 0.1160	*μ* = 0.0028, *σ* = 0.3860, *μ*_|*ϕ**−*ϕ*|_ = 0.2298	*μ* = 0.0498, *σ* = 0.2039, *μ*_|*ϕ**−*ϕ*|_ = 0.3451

## Data Availability

Data are contained within the article and Appendix A.

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
