# Peer review of "A Best-Fitting B-Spline Neural Network Approach to the Prediction of Advection–Diffusion Physical Fields with Absorption and Source Terms"

_entropy, 2024, doi:10.3390/e26070577_

Round 1

Reviewer 1 Report

Comments and Suggestions for Authors

In this work,  a two-dimensional steady-state field prediction approach that combines B-spline functions and a fully connected neural network is proposed. Numerical examples compared with GAN and PINN shows that the proposed method has high prediction accuracy, less demand for training data and high training efficiency. The result is interesting and the paper is well written. I recommond to accept it.

Author Response

Thank you, please see the attachment.

Reviewer 2 Report

Comments and Suggestions for Authors

This paper presents a neural network architecture for advection-diffusion equations that is based on a b-spline basis expansion of the unknown, where the coefficients are returned from a neural network.

It is unclear to me what is the novelty in the manuscript, or what is the research question that the authors are addressing.

1. The authors seem to overlook a large body of research in the area of operator learning which also tend to express solutions of PDEs in terms of basis functions (see e.g. DeepONets (and its many variants such as physics informed deeponets), VarMiONs, PCA-nets,  BelNets ...etc). A special case of many of these networks coincides with the presented architecture, namely when the so-called trunk-networks are fixed to some b-spline basis. What does the proposed architecture offer in comparison to such networks? This needs to be addressed in future versions of this manuscript.

2. This paper also lacks a mathematical error analysis. This can provide a theoretical justification/advantage of using the proposed network architecture. Perhaps it may be an option to see how the current architecture fits within the error analysis of the aforementioned networks? 

3. The paper does not provide details of the training procedure. Its unclear how the hyper-parameters were tuned, and how was the training and testing datasets generated. Its customary to provide links to the code repository with the paper as well. Perhaps the authors could consider that for the next version of the manuscript.

Unfortunately, based on these comments i can't recommend publishing this paper in its current form, as i find it lacking in several areas, including background theory, methodology and numerical results. 

Author Response

Thank you, please see the attachment.

Reviewer 3 Report

Comments and Suggestions for Authors

The manuscript presents interesting results regarding the numerical study of advection-diffusion physical fields. My major concerns and recommendations are:

1) The authors should provide a comparison of the proposed approach to an analytical solution. The book fo Crank on diffusion (CRANK, J. The Mathematics of Diffusion. 2nd. Ed. Oxford University Press: Oxford, 1980) presents plenty of analytical results that could have been used. 

2) A cylindrical-geometry-based study should also be presented.

Author Response

Thank you, please see the attachment.

Round 2

Reviewer 3 Report

Comments and Suggestions for Authors

The revised version can be accepted.